# Actions for Monitoring the *Gonipterus* Pest in Eucalyptus on the Cantabrian Coast

Esperanza Ayuga-Téllez [1], Alberto García-Iruela [2], José Causí Rielo [3] and Concepción González-García [4],*

1 Buildings, Infrastructures and Projects for Rural and Environmental Engineering (BIPREE), Universidad Politécnica de Madrid, Camino de las Moreras s/n, 28040 Madrid, Spain; esperanza.ayuga@upm.es
2 Tecnología de la Madera y el Corcho, Universidad Politécnica de Madrid, Ciudad Universitaria s/n, 28040 Madrid, Spain; agiruela17@gmail.com
3 Forestry Director at ASPAPEL, Av. de Baviera, 15 Bajo, 28028 Madrid, Spain; j.causi@aspapel.es
4 Methods & Technologies for Sustainable Management Research Group (MTSM), Technical School of Forest Engineering, Universidad Politécnica de Madrid, Camino de las Moreras s/n, 28040 Madrid, Spain
* Correspondence: concepcion.gonzalez@upm.es

**Abstract:** Forests are a natural resource of great importance for sustainable development. They represent the primary use of Spanish territory and cover 36% of its area. Eucalyptus shrublands are the most productive, particularly on the Cantabrian coast, occupying a total area of 406,566 ha. Since 1991, some of these shrublands have been affected by the eucalyptus snout beetle (*Gonipterus platensis*), a coleoptera (weevil) from the Curculionidae family that feeds on eucalyptus leaves and produces significant damage. The innovation project of the Supra-regional Operational Health Group on *Gonipterus* in *Eucalyptus* was developed (2019–2020) to establish a global approach to the serious problem it causes in Asturias and Galician *Eucalyptus* stands. A group of experts devised two action protocols to unify the methods and variables measured in the field: a protocol for measuring and estimating damage (degree of defoliation) and a parasitism protocol to establish actions to monitor the degree of parasitism (collection of oothecae, management of the sample, laboratory procedure). In the results, in addition to establishing the sampling protocol, an analysis of the data (from 2017 to spring 2020) provided by the different administrations of the Autonomous Communities studied has been carried out. The data analysis reveals an improvement in the impact of the damage on the Cantabrian coast (29.8% reduction in damage in Galicia and 14.7% in Asturias). In Galicia, the number of adult insects decreased from 2017 to 2019, increasing in the spring (from April to June) of 2020 above the mean values of previous years in that period. The number of larvae in the different larval stages showed similar development in all cases. The mean larvae (in their different stages) and mean oothecae showed a significant decline in the year 2018 compared to the spring of 2017, with an upturn in 2019 and again similar values to 2018 in the spring of 2020. In Asturias, similar mean values of the order of 0.5 insects per plot on dates (May–June) in spring were observed in 2019. While in 2020, a progressive increase could be seen in the mean number of insects throughout March, up to 1.9 insects per plot. Results of research on the biological treatment of parasitisation of oothecae with *A.nitens* were also collected to adjust the number of oothecae per bag deposited in the field and the number of *Anaphes* released per ha. Based on the field observations, the appropriate release time was determined to succeed in controlling the *Gonipterus* population.

**Keywords:** eucalyptus snout beetle; forest health; forest damage; biological control; *Eucalyptus globulus*; monitoring protocol

## 1. Introduction

Forests are a natural resource of great importance for sustainable development. They represent the primary land use in Spain, with 55% of forested land, and forests covering 36% of the area of Spain [1,2].

Eucalyptus shrublands are among the most productive of these forests, particularly on the Cantabrian coast. Eucalyptus wood is used in cosmetics, household utensils, textiles, manufacture of boards and primarily for the production of cellulose and paper (writing paper, toilet paper, sanitary products and others). Spain and Portugal cover 40% of the demand for this cellulose in the European Union [3]. The area covered by eucalyptus occupies 406,566 ha on the Cantabrian coast and Galicia [4,5]:

- Galicia: 287,983 ha;
- Asturias: 60,311 ha;
- Cantabria: 39,522 ha;
- Basque Country: 18,750 ha.

The largest extension corresponds to Galicia and Asturias, which together represent 85.67% of the total area of eucalyptus on the Cantabrian coast.

Some of these plantations are affected by damage caused by a defoliating insect, the eucalyptus snout beetle (*Gonipterus platensis*, Marelli, 1926), which first appeared in 1991 in the locality of Lourizán in Pontevedra (Galicia, Spain) [6]. The insect responsible for the damage is a coleoptera from the Curculionidae family that feeds on eucalyptus leaves. The loss of leaves affects the growth of the tree and thus decreases $CO_2$ fixing by the forest stands. *Gonipterus platensis* reduces the normal growth of eucalyptus by 20–25%, with an annual loss of between 800,000 and 1,200,000 tons of timber and an economic impact of approximately EUR 235 million per year. It also leads to a loss of employment estimated at 42 jobs/year [7].

*G. platensis* has its greatest distribution outside its area of native distribution. Originally from Tasmania, this species was accidentally introduced in western Australia, New Zealand, Europe (Portugal and Spain), South America (Argentina, Brazil and Chile) and the USA (California and Hawaii) [8]. On the Iberian Peninsula, it spread from Lourizán to the provinces [9], and by 1997 had colonised practically the whole of Galicia. The favourable climate conditions, coupled with the absence of natural enemies, favoured the dispersion of this curculionid, which subsequently appeared in Navia, Asturias, in 1994; northern Portugal (1995); Bakio, Vizcaia, in the Basque Country (1997); the Canary Islands (1999); and Cantabria (1999) [10].

The life cycle of *G. platensis* [11] comprises two generations a year [12,13]; adults are able to survive between six months and one year, and sometimes even two years. Both the adults and the larval states cause damage due to defoliation. The first adults emerge in mid-February and early March and can feed through the winter in warm climates. They feed on young leaves and begin to lay eggs after approximately one month. The four larval states (L1, L2, L3, L4) develop in approximately one month. The L1 larvae hatch 10–15 days after laying (L1 and L2 feed on the epidermis), while the L4 larvae fall to the ground, dig down 10–15 cm and build an ovoidal cell in which pupation takes place [14,15]. This period lasts between 30 and 50 days (corresponding to a period of summer semi-latency, when practically no activity is observed), after which the adults emerge, giving rise to the second generation.

The biological control of the eucalyptus snout beetle involves the use of a natural parasite (*Anaphes nitens*, Girault, 1928) of *Gonipterus* eggs. *Anaphes nitens*, a Hymenoptera in the Mymaridae family, is a natural and specific parasite of *Gonipterus* eggs, which grows to barely 1 mm in length as an adult. After mating, the females lay their eggs in the interior of the *Gonipterus* eggs, which are then parasitised. The larvae of *A. nitens* remain inside the egg and feed on its contents, thus preventing the development of the larvae of *G. platensis*. Once it has reached adulthood, *A. nitens* emerges in search of new oothecae to parasitise. Numerous studies have been carried out on the successful parasitisation of *A. nitens* in *G. platensis* on the Iberian Peninsula [16–19]. Despite the favourable results obtained with *A. nitens* in many regions, this control has not been successful everywhere, particularly in some regions of South America [20], Western Australia [21] and southwest Europe [17,22,23]. The differing climate requirements of *A. nitens* and *G. platensis* and the asynchrony between the laying times of the beetle and the parasite may explain the lack of efficacy of the biological

control [17,21,24]. In cold regions, leaf-flushing by eucalyptus trees is inhibited by low temperatures during the winter months, which reduces the number of suitable sites for the females of *G. platensis* to lay their eggs and consequently causes a decline in the number of hosts available for *A. nitens* [21,24,25]; this, in turn, leads to a decrease in the populations of *A. nitens* in winter. At the end of winter/early spring, when *G. platensis* begins to lay more eggs, *A. nitens* cannot respond in sufficient numbers and suffers high rates of mortality. Although the rates of parasitism at the end of the spring may be over 90%, the snout beetle larvae that escape parasitism early in the season have already caused defoliation [17,21,22,24,25]. As successful control has not been achieved with *A. nitens* in several major eucalyptus-producing regions, classic biological control (CBC) with other natural enemies must be considered. Several natural enemies of *Gonipterus* spp. have been reported from Australia. Some of these are described in [26], namely two wasps, *Euderus* sp. (Hymenoptera: Eulophidae) and *Centrodora* sp. (Hymenoptera: Aphelinidae). In Tasmania, [24] reported the presence of the larval parasitoids *Oxyserphus turneri* (Dodd) (Hymenoptera: Proctotrupidae), *Apanteles* sp. (Hymenoptera: Braconidae) and an unidentified tachinid. In 2011, the larval parasitoid of *Gonipterus* spp. *Entedon magnificus* (Girault and Dodd) (Hymenoptera: Eulophidae) was collected in Tasmania and has been demonstrated to successfully parasitise *G. platensis* [20]. The egg parasitoids *Anaphes tasmaniae* Huber and Prinsloo, *Anaphes inexpectatus* Huber and Prinsloo and *Centrodora damoni* (Girault) (Hymenoptera: Aphelinidae) are known to be present in Tasmania [24,27,28]. *Centrodora damoni* has also been reported in Queensland and Canberra [28]. *Podisus nigrispinus* (Hemiptera: Pentatomidae) is a predator of larvae and adults of *G. platensis* [29].

Other options are being studied in the laboratory, including the use of compounds that could serve as traps. "Three compounds attracted virgin females or both sexes: camphene, (+)-α-pinene and 2-phenylethanol are potential candidates for application in integrated pest management approaches. Furthermore, the two allomones may have potential use for developing push–pull strategies" [30]. *Steinernema diaprepesi* was also identified as the entomopathogenic nematode that can control the pest due to the mutual association with bacteria that reproduce and kill the pupae of *G. platensis* through septicaemia [31]. Two strains of entomopathogenic fungi that are especially lethal for the adults of *G. platensis* [32] have been found in Brazil. On the Iberian Peninsula, studies have been conducted on the viability of *A. inexpectatus* in the parasitisation of *G. platensis* [26,33,34] and on its interrelation with *A. nitens* [35].

For many years on the Iberian Peninsula, the means of monitoring pests and issuing an early alert were through terrestrial prospects using protocols designed in each country [36,37] or autonomous region [38,39] or through private projects [40], which measured different parameters, making it impossible to compare the results. Other works have used drones to monitor pests [41], including the GOSSGE project [7], which used drones for aerial prospection in a small area of Asturias. A recent study [42] analysed the case of *G. platensis* in Spanish eucalyptus plantations. These authors combine the data set of the Spanish national plots belonging to the European transnational sampling network on the state of forests in Europe (International Cooperative Programme on Assessment and Monitoring of Air Pollution Effects on Forests, http://www.icp-forests.net (accessed on 22 May 2022), ICP Forest Level I, 16 × 16 km grid), together with regional plots (mainly from Galicia) in which *Eucalyptus* spp. are present, measured using similar field protocols.

The combined plans and programmes established to combat the eucalyptus defoliator are the first case of a large-scale classic biological control at the forest level carried out in Spain [43].

The aim of this work is to evaluate the survey of the damage conducted by the Autonomous Regions (AR) and to propose a general methodology that can be applied anywhere. The results of the 2018–2020 biological control campaign were also analysed. The research work is a part of the project of the Supra-regional Operational Health Group on *Gonipterus* in eucalyptus (GOSSGE) [7,44], whose main function is to ensure the sustainability of the eucalyptus stands in northern Spain.

## 2. Materials and Methods

### 2.1. Study Variables

The variable used to assess the survey is the degree of defoliation. Defoliation is a basic parameter for quantifying the apparent state of health of a tree stand and is defined as the loss or lack of development of leaves or needles suffered by the tree in the evaluable part of its crown compared to that of the ideal reference tree in the area. The percentage of defoliation is calculated in the upper third of the crown [7]. Degrees of defoliation comprise ordinal variables that take values from 0 to 4, where 1 corresponds to damage of 10% to 25% and a degree of 2–3 from 26% to 90% [45].

The biological control was conducted with bags of oothecae parasitised by *A. nitens*. The variables used to monitor the control were the number of bags deposited with parasitised oothecae, the number of oothecae per bag and the number of insects and larvae collected.

### 2.2. Study Area

The two ARs with the largest forested area covered by eucalyptus are Galicia and Asturias. The affected areas monitored on dates prior to this work [46] and which serve as a basis for this project are shown in Table 1.

**Table 1.** Areas of eucalyptus and the affected part according to the degree of defoliation (2013).

| Autonomous Region | Area of Eucalyptus CCC $\geq$ 70% | Affected Area Degree 1 | Affected Area Degree 2–3 |
|---|---|---|---|
| GALICIA | 163,664 | 96,493 | 17,457 |
| ASTURIAS | 42,345 | 19,412 | 5950 |
| **TOTAL** | 206,009 | 115,905 | 23,407 |

CCC = closed canopy cover.

### 2.3. Information Used

Statistical analyses of the variables described in Section 2.1 were performed with the following information:

- data provided by the authorities in the respective AR (Xunta de Galicia and Principado de Asturias);
- data supplied by the Spanish Association of Pulp, Paper and Cardboard Manufacturers (ASPAPEL);
- data provided by ADRA Engineering and Environmental Management (SLP) on the two regions;
- data from additional surveys by the Forest Owners Association of Asturias (PROFOAS);
- data provided by the Spanish Confederation of Forestry Producer Organisations (COSE) for each region.

A total of 10 trees were assessed in each sampling plot. To correctly observe the tree, the evaluators stood with the sun behind them, at a minimum distance equal to the height of the tree to be evaluated and at the same level or slightly higher. In the case of slopes, the assessment was conducted from a higher level in order to obtain the most complete observation possible of the tree crown [38,40]. For its assessment, it was recommended to divide the crown into different parts of a similar size and to assess each one individually and register the mean of the estimates [40]. Additionally, in the case of the data collection in Galicia, the count included the number of adults, oothecae and larvae on twigs with a length of 10 cm selected from the upper third of the crown [38].

### 2.4. Focus Groups

Group meetings or focus groups, one of the most widely used qualitative methodologies, were created to generate the action protocols [47]. These are meetings with participants with a specific profile selected to cover each research need (in this case, professionals or

experts in the forestry sector). The group meetings followed a structured approach: the qualitative technician (the ASPAPEL representative) prepared a guideline for the order of presentation of the aspects needing to be taken into account for the survey and biological control of the damage with the parasitoid. Four virtual meetings were held with an approximate duration of between 2 and 2.5 h. Eight representatives from the following organisations took part: ASPAPEL, COSE, Madrid Polytechnic University (UPM), the National Cellulose Company (ENCE) and representatives of the AR of Asturias and Galicia.

The degree of defoliation was assessed with a model of the state of the stand based on the comparison of photos [48].

### 2.5. Statistical Analysis

Descriptive methods were used. Mathematical expressions for stratified random sampling were applied to calculate the correct sample size [49], with the AR as strata and with levels of confidence of 90% and 95%.

The minimum number of plots to be assessed in each region was optimised in order to obtain the closest possible values to the state of the damage caused by the pest. This was conducted by calculating the optimum number of plots ($n_0$) per AR with an area of eucalyptus affected by *Gonipterus* in 2019, a 10% error of estimation and a confidence level of 95%, considering the population as the whole area of the eucalyptus forest. Thus 153 plots were obtained in Galicia and 100 in Asturias.

These values were adjusted to the eucalyptus area in proportion to the points in the sampling grids obtained from the ASPAPEL survey reports (2013) for each AR. Following the indications in these reports, we considered the possible sampling points (N) shown in Table 2. The sample size [50] for the survey in each region was calculated with the Equation (1):

$$n = \frac{n_0}{1 + \frac{n_0}{N}},$$ (1)

where $n$ is the number of sampling points to be surveyed, $n_0$ is the optimum sample size calculated for an infinite population and $N$ is the number of points in the sampling grid obtained according to the eucalyptus area proposed in ASPAPEL's sampling protocol [46] (Figure 1).

**Table 2.** Optimum sampling points by Autonomous Region.

| Autonomous Region | $n_0$ | N | n |
|---|---|---|---|
| Galicia | 153 | 174 | 82 |
| Asturias | 100 | 90 | 48 |

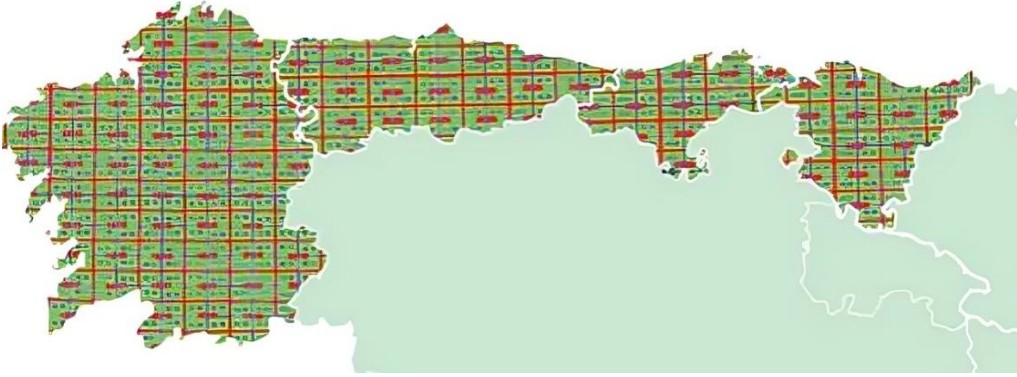

**Figure 1.** Map of the Cantabrian coast with ASPAPEL's sampling grid.

The sampling points were randomly selected.

The data were processed with an Excel spreadsheet and the Statgraphics Centurion XVIII statistical software. These tools were used to obtain summary statistics and graphics, and Fisher's LSD method was applied to create confidence intervals for factor-level means.

## 3. Results

### *3.1. Results of Focus Group*

This research required a survey protocol as the results from the two ARs had to be standardised and compared. The aim was to obtain a faster, more economical, objective and comprehensive evaluation of the stand.

Group meetings were held to devise a protocol for visiting selected plots and the methodology to obtain and archive the various data collected in the observations, measurements and counts made in each plot (see Sections 2.1 and 2.3). The value (*n*) obtained in Table 2 is the number of plots in which the protocol would be applied.

The variables to be measured in the field were also unified in order to establish the presence or absence of *Gonipterus* at each point, the development of its biological cycle and the degree of defoliation. For the biological control, the collection of oothecae was determined in each plot to assess the degree of parasitisation by *A. nitens* in the laboratory.

3.1.1. Sample Selection

The criteria for selecting the eucalyptus stands to establish the survey plots were based on the following parameters:

- Location: eucalyptus stands that cover the spatial variation of the *Eucalyptus globulus* species in the study region;
- Level of impact: stands with different degrees of defoliation caused by *G. platensis;*
- Age of the stands: young eucalyptus stands between three and six years old;
- Accessibility: stands with ease of access for monitoring.

The sampling plots (Table 2) were located in the centre of the plot at randomly selected sampling points from between the vertices of the proposed grid (Figure 1); 10 trees for the sampling had to be located in the area around this point.

The area was marked on the plot or obtained from the sampling point using the necessary distance to locate the 10 trees.

Each tree in the sample was identified with a number and code on the corresponding Excel sheet.

The damage to the plots was assessed between March and October (with a break in July and August), as these are the months when the pest is most active (Figure 2). The start of the evaluation varied depending on the climate conditions in each specific year and geographic area and in the case of an early alert to the detection of the damage.

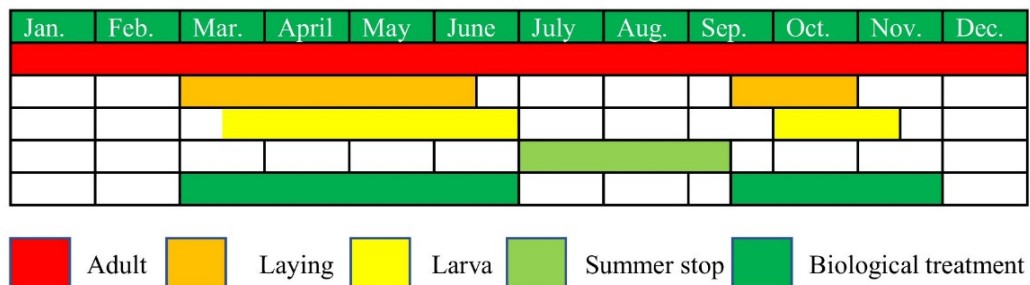

**Figure 2.** Diagram of the snout beetle life cycle.

The sampling took place once a month, with a minimum of two annual surveys, preferably in March or April and May or June, based on the damage level in certain years and the geographic features of the sampling area.

### 3.1.2. Evaluation of the Degree of Defoliation

It is important to note that the percentage of defoliation is calculated in the upper third of the crown, as shown in Figure 3.

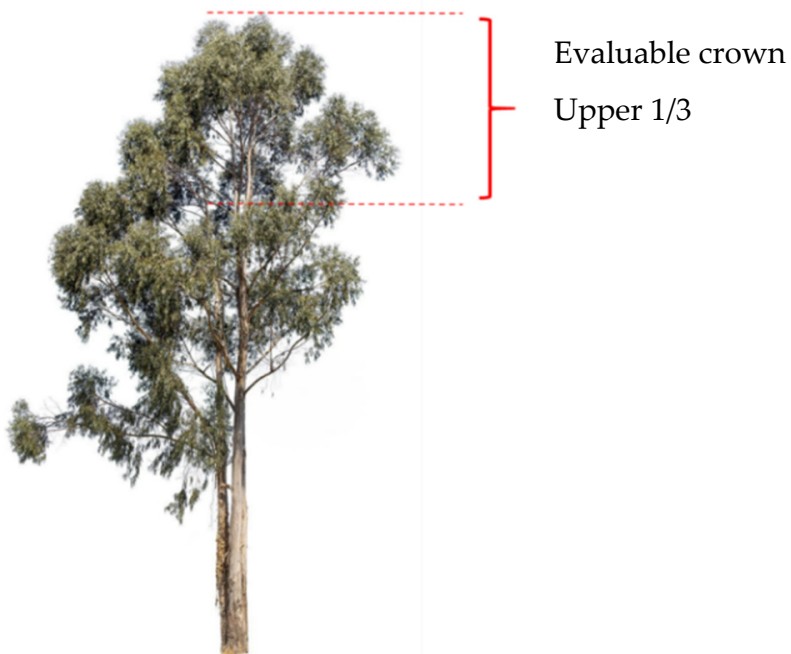

**Figure 3.** Diagram of the upper third of the crown of *Eucalyptus globulus* where the degree of defoliation is assessed.

The degree of defoliation was evaluated by the observer, where possible, the same person, according to the five levels or degrees of defoliation agreed in the group meetings (Table 3 and Figure 4).

It is useful to show the following in the field data collection: the damage level, the stage in the pest cycle and the abundance of adults, larvae and oothecae. It is also important to note the presence or absence of other pests or diseases on the stand.

### 3.1.3. Oothecae Collection

The aim of collecting the oothecae is to observe the levels of parasitisation by *A. nitens* in the eggs of *G. platensis*. Whenever possible, 10 fresh oothecae were collected from the 10 trees on each visit or survey for their subsequent study in the laboratory.

The following information must be documented on each visit:

- Date (xx/xx/xxxx);
- Author (first name, surname and company or organisation);
- Plot data;
- Plot no. and/or code;
- Visit no.;
- Area;
- Location (shrubland, town, district, province);
- UTM coordinates (X, Y) and reference system;
- Altitude (m);
- Species;
- Age of the shoot and/or stand (years);
- No. of trees surveyed;
- Degree of defoliation for each individual evaluated (see Figure 4);
- State of the damage;
- No. of fresh oothecae collected from the selected trees;

- Presence of *Mycosphaerella* spp.;
- Presence of *Ctenarytaina* spp.

**Table 3.** Degrees of defoliation according to the GOSSGE protocol.

| Degree of Defoliation | Tree Crown Damage |
|:---:|:---:|
| 0 | 0–10% |
| 1 | $\geq -25\%$ |
| 2.1 | 26–45% |
| 2.2 | 46–60% |
| 3 | 61–90% |
| 4 | >90% |

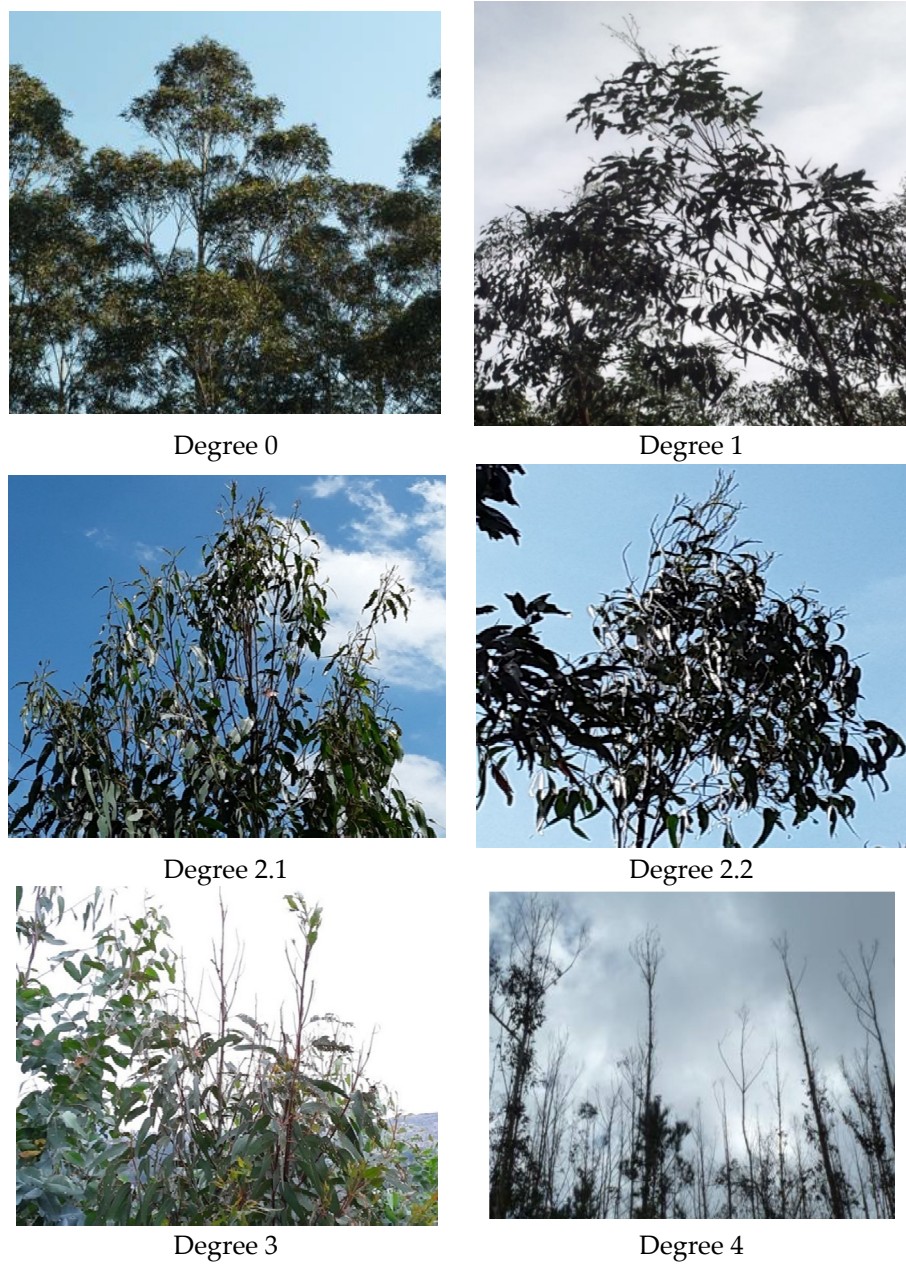

Degree 0

Degree 1

Degree 2.1

Degree 2.2

Degree 3

Degree 4

**Figure 4.** Photographic record of the defoliation degrees according to the GOSSGE protocol.

### 3.2. Analysis of Survey Data in Galicia and Asturias

The number of plots surveyed in both regions was modified according to the data available on the degree of annual defoliation in each one. The annual data are therefore shown separately.

### 3.2.1. Galicia

In this region, information from the Xunta was available from 2017 to 2019, in addition to data for spring 2020, when the Xunta had detailed information on the eucalyptus damage with a high number of sampling plots in Pontevedra, A Coruña and Lugo. In Ourense, there was a very low incidence of damage and a smaller area of eucalyptus woods, so very few samplings were conducted.

A summary of the data for the degree of defoliation was obtained for the whole of Galicia (Table 4).

**Table 4.** Percentage of area with degrees of defoliation according to the GOSSGE protocol for the three provinces in Galicia with damage in 2019 and spring 2020.

| Degree of Defoliation | A Coruña (%) | | Lugo (%) | | Pontevedra (%) | | Total (%) | |
|---|---|---|---|---|---|---|---|---|
| | 2019 | Spring 2020 | 2019 | Spring 2020 | 2019 | Spring 2020 | 2019 | Spring 2020 |
| 0 | 88.30 | 96.08 | 93.14 | 93.75 | 0.33 | 11.40 | 75 | 84.67 |
| 1 | 11.66 | 1.38 | 6.86 | 5.43 | 37.90 | 24.68 | 14.8 | 5.49 |
| 2.1 and 2.2 | 0.04 | 1.80 | 0.00 | 0.82 | 61.78 | 63.41 | 10.2 | 9.40 |
| 3 and 4 | 0.00 | 0.74 | 0.00 | 0.00 | 0.00 | 0.51 | 0 | 0.44 |

The evolution in each year is shown in Figure 5, according to the dates of each survey or the survey number, as the dates in each year do not always coincide, and the number of surveys may vary from year to year.

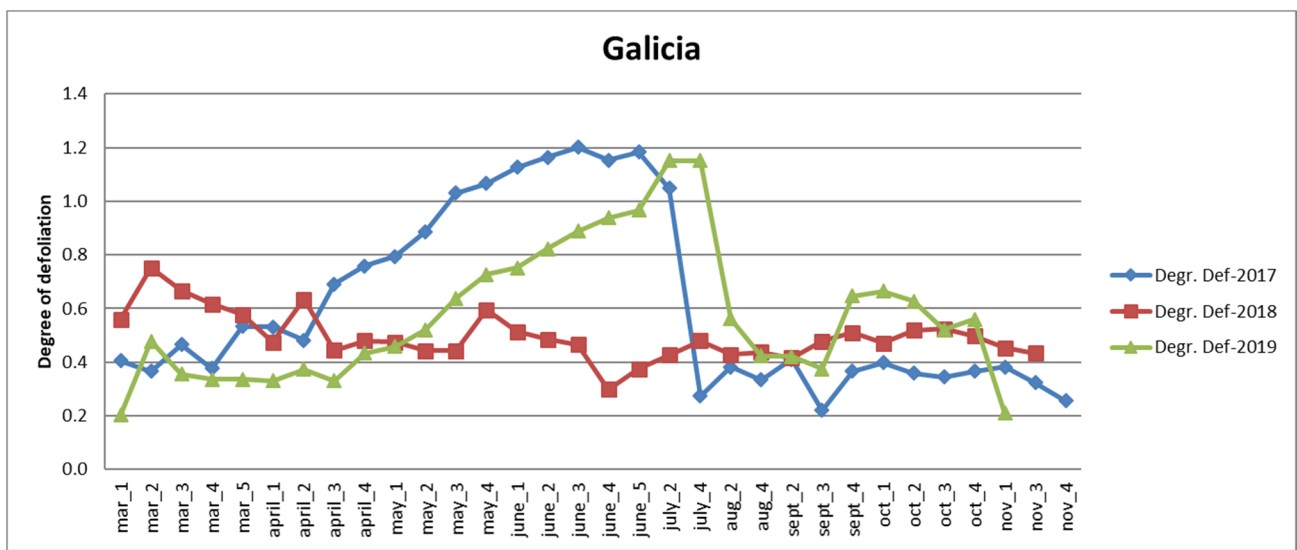

**Figure 5.** Evolution of the degree of defoliation according to the week of the sampling month (visit).

From the 2017–2019 analysis, it can be concluded that the damage declined from 2017 to 2018, then rose again in 2019 but without reaching the levels of 2017. In 2020 it should be noted that data are only available for spring until the end of May.

Of the three provinces, Pontevedra was the most affected and Ourense the least (the values in all the sampling points were 0 degrees of defoliation). Table 4 show the percentage of the estimated area in each province where significant damage was detected in 2019.

For the data for Galicia for 2020, the graph (Figure 6) show the number of plots according to the survey or week of the visit (from March to May 2020) and a summary of

the values (mean, minimum and maximum) for the degree of defoliation in the plots for the whole region.

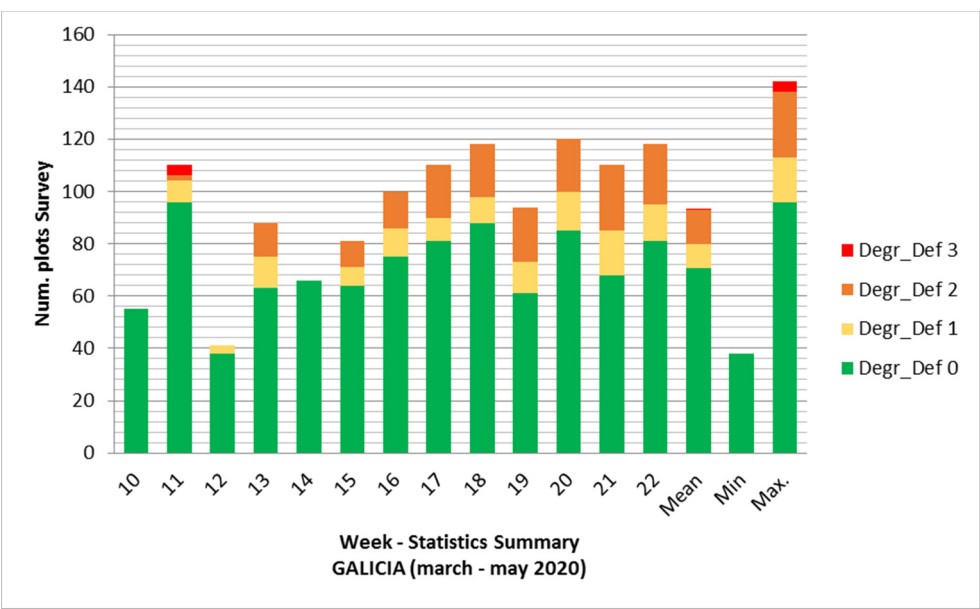

**Figure 6.** Number of plots sampled in Galicia in 2020 by week and degree of defoliation.

The total number of adults was also evaluated for each year according to the survey (Figure 7). In this case, the dates are shown by the number of the week in the month, as they are not the same each year.

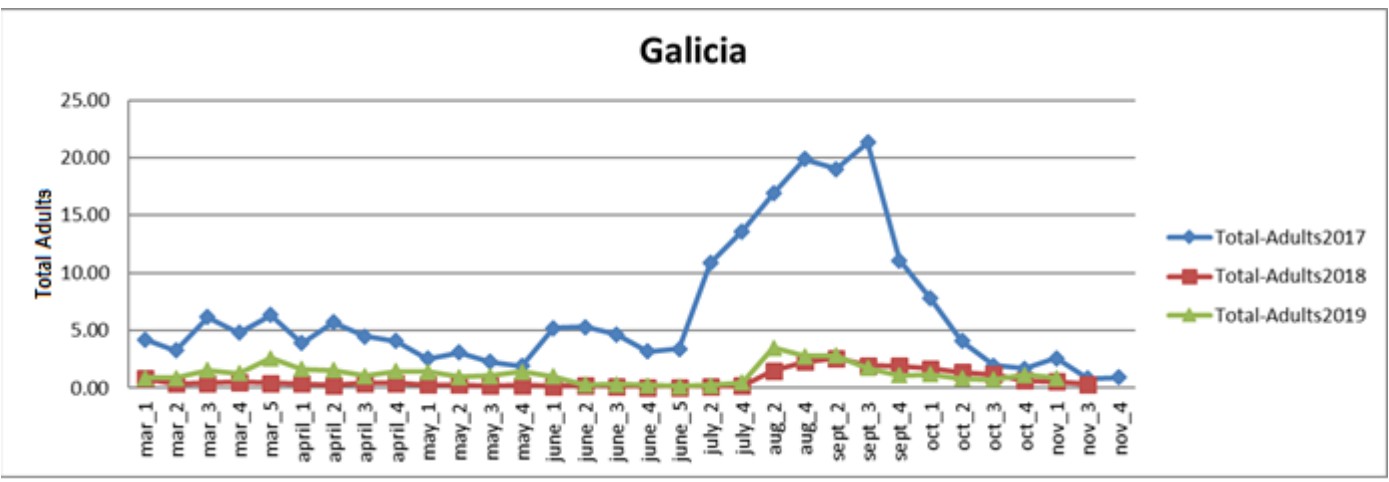

**Figure 7.** Mean values for total adults according to the week of the sampling month (visit).

The insect infestation had a greater presence in 2017, with a high abundance of adults from July to September, and lower in subsequent years. The number of adults in 2018 and 2019 was very similar. The mean value of total adults per plot evaluated in spring 2020 was 9, higher than the values for spring in previous years.

The number of larvae in the different larval stages showed a similar development in all cases (Figures 8–11). Low values increased in mid-April and then decreased again in late July, except in the first larval stage when the decrease occurred in mid-June. The values detected in spring 2018 show a significant decline in the abundance of larvae compared to the spring of 2017, with an upturn in 2019.

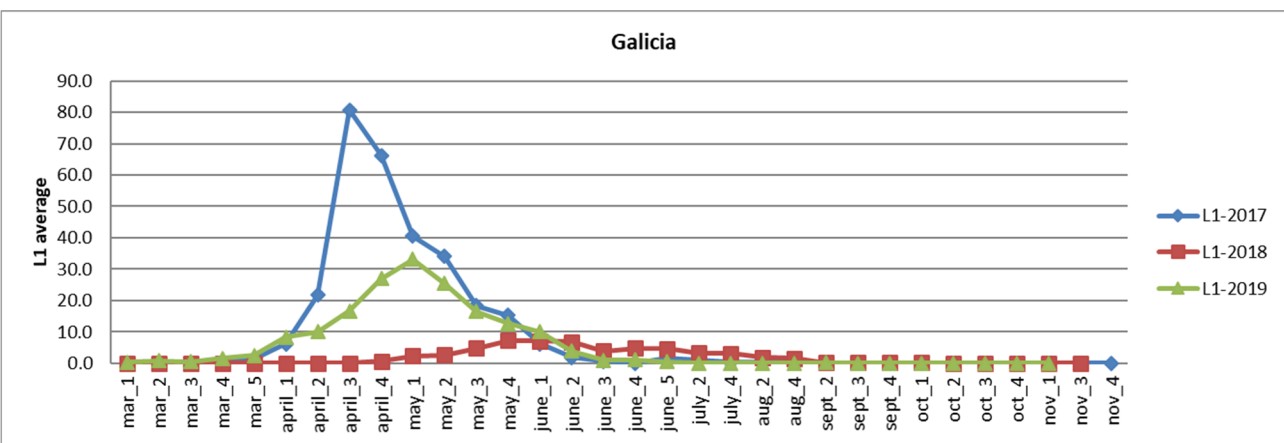

**Figure 8.** Evolution of the average number of larvae in stage L1, according to the week of the survey month.

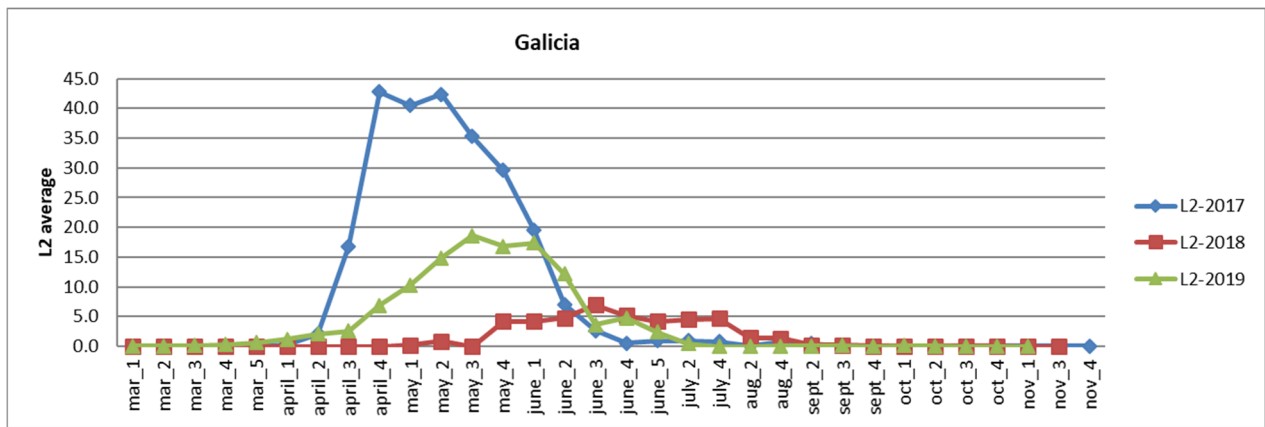

**Figure 9.** Evolution of the average number of larvae in stage L2, according to the week of the survey month.

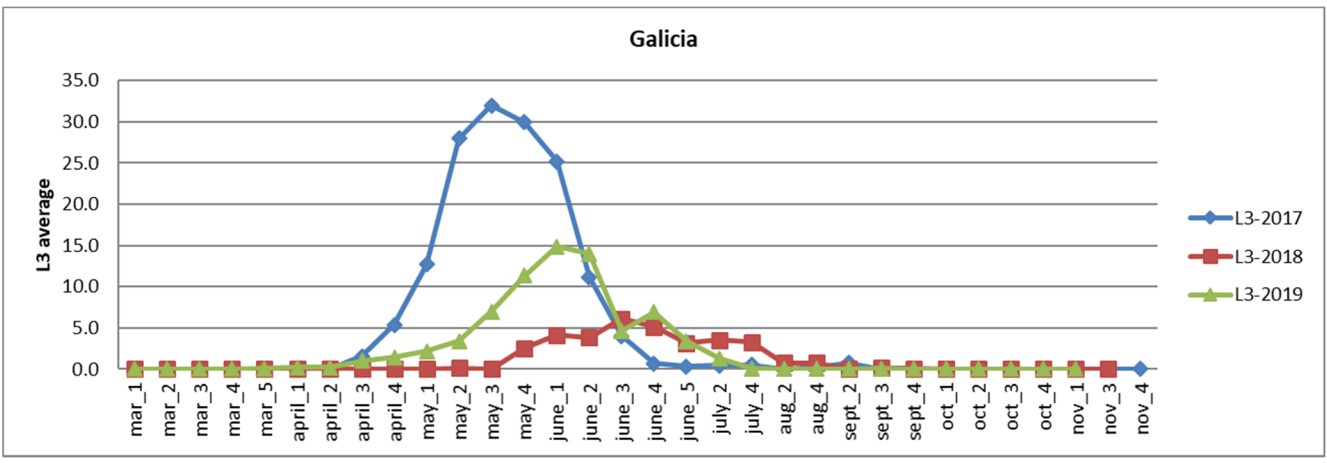

**Figure 10.** Evolution of the average number of larvae in stage L3, according to the week of the survey month.

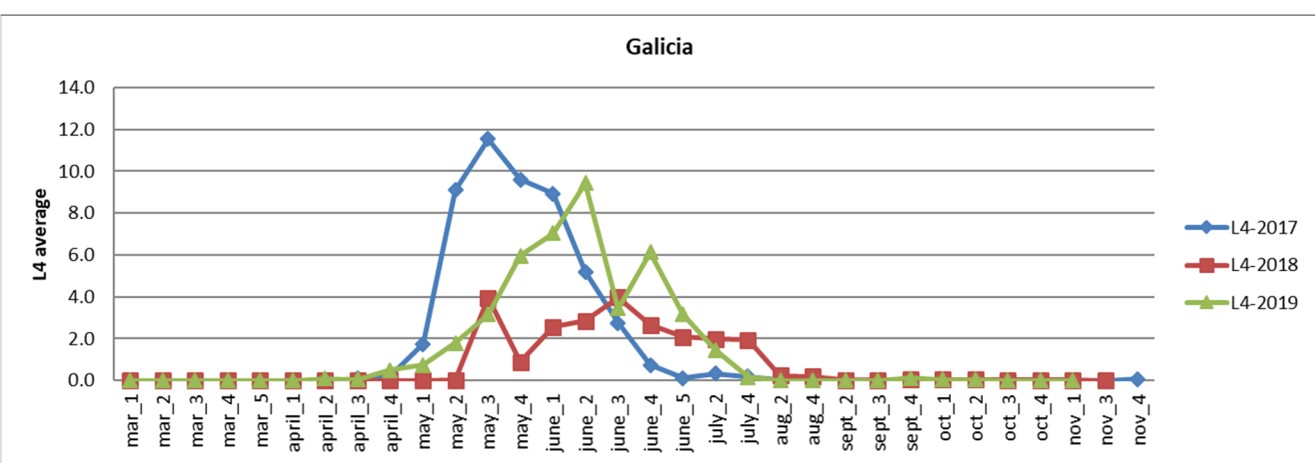

**Figure 11.** Evolution of the average number of larvae in stage L4, according to the week of the survey month.

Figure 12 show the evolution of the variable mean values for the number of oothecae collected by date of visit or survey, according to data from the Xunta data. The mean values were obtained because the total number of oothecae per survey corresponded to counts from a different number of plots each time.

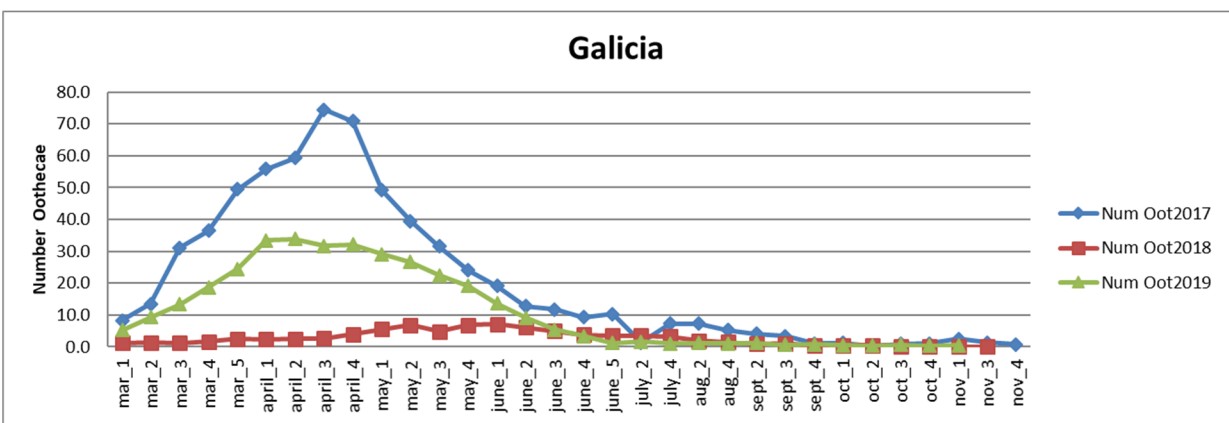

**Figure 12.** Number of oothecae according to the week in the sampling month (survey).

As Figure 12 shows, the number of oothecae was high in 2017, then decreased in 2018 and rose again in 2019, without reaching the levels of 2018. In spring 2020, the mean number was 6 oothecae per plot, similar to the values for 2018.

3.2.2. Asturias

Data were available for 87 plots in this region for spring 2019. Only data from ADRA were available for autumn 2019 to be able to continue analysing the same variables as in spring, considering that the sample size increased to 89 plots in spring 2020. Seven visits were made to plots in March, and a total of 202 plots were sampled. The data were provided by ADRA.

Data on the degree of defoliation are shown in Table 5. The average degree of defoliation in spring 2019 was 0.97, rising to 1.2 in autumn 2019, while the figure was 1.01 in spring 2020. The mean values remained fairly stable in the three periods.

**Table 5.** Degrees of defoliation according to the GOSSGE protocol and percentages of affected area for the three data collection periods.

| Degree of Defoliation | Spring 2019 | Autumn 2019 | Spring 2020 |
|:---:|:---:|:---:|:---:|
| 0 | 11.24% | 28.87% | 24.75% |
| 1 | 57.30% | 41.24% | 52.97% |
| 2.1 | 26.97% | 27.84% | 18.81% |
| 2.2 | 4.49% | 2.06% | 2.97% |
| 3 and 4 | 0.00% | 0.00% | 0.50% |

Table 5 show the variation in the percentage of area affected with each degree of defoliation between autumn 2019 and spring 2020. It can be seen that the area with high levels of defoliation (2.1 and 2.2) has decreased, the area with level 0 has increased and the area with level 1 has increased slightly.

No insects were observed in 81.2% of the plots, and 1200 were estimated in 12.9% of plots. For the temporal monitoring of the pest, mean values for the number of insects per plot were observed in 2019 (Figure 13a) and 2020 (Figure 13b).

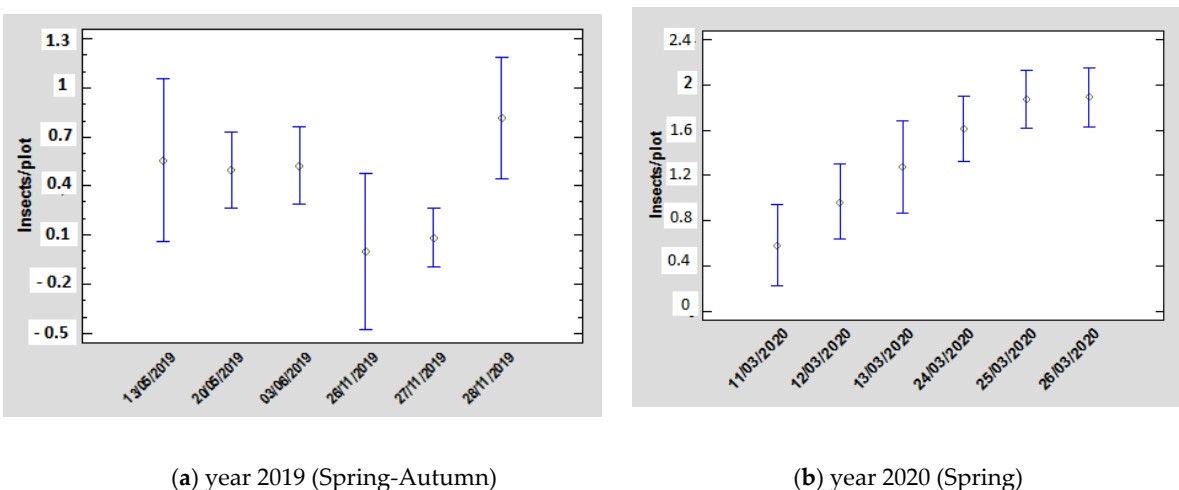

(**a**) year 2019 (Spring-Autumn)          (**b**) year 2020 (Spring)

**Figure 13.** (**a**,**b**) Mean number of insects per plot with confidence intervals up to 95%, according to the monitoring date.

Similar mean values of the order of 0.5 insects per plot on dates in spring were observed in 2019. The observations in late November vary widely from one plot to another, with a higher value close to 1 on 28 November. In 2020, a progressive increase could be seen in the mean number of insects throughout March (Figure 13b), up to 1.9 insects per plot. The data from May to June 2019 show stable means compared to March 2020. Observations for April are required to improve the knowledge of the development of the insect's cycle. In Figure 13b, a very long interval can be seen on 19 March 2020 due to the low number of plots sampled, n = 2.

In spring 2019, no oothecae were collected in 87.94% of the plots, and a maximum number of 10 oothecae in 2.25% of the plots. The mean value was 0.78 oothecae per plot. In autumn 2019 and March 2020, no oothecae were collected on the date the samples were taken.

*3.3. Biological Control*

For the biological control of the pest, the Xunta specifies placing bags in the treatment area at a rate of one bag with 50 oothecae every 2 ha, while ADRA recommends one bag every 6 ha with a different number of oothecae in each bag. Between 2019 and 2020, 257,403 parasitised oothecae were placed in 8004 ha distributed along the Cantabrian coast, with parasitism rates (no. *Anaphes*/ootheca) of 2.95 in 2019 and 4.15 in 2020 [7].



Research plots both with and without biological treatment were used in the survey, and the biological control was observed to work with damage of less than 43% in the upper part of the crown (a maximum degree of defoliation of 2.1) [7]. In treated areas and with an increase in parasitisation of up to or over 50%, there is a 75% likelihood of suffering damage below the admissible limit of 43% in the upper third of the crown. It was also concluded that the released dose of 100 *Anaphes*/ha was the most adequate [7]. Similar results were achieved with a lower dose of parasitoid release than used previously [7].

Based on the field observations, the appropriate release time was determined to be right at the start of the appearance of the first oothecae. If the release takes place at the ideal time, the parasitism rate rises sharply, and the damage is reduced as the rapid parasitisation of the oothecae succeeds in controlling the *Gonipterus* population.

It was also observed that the *Anaphes* expand in all directions from the release point and use the prevailing wind for their displacements and that the oothecae are parasitised with the same intensity at different distances from the release point.

## 4. Discussion

Commercial forest plantations require an assessment of the state of health of the tree stand in regard to the presence of insect pests and disease [50]. The eucalyptus weevil has for many years been controlled through the use of its parasitoid [51] with different results [52]. In spite of this, very few studies have evaluated its effects in eucalyptus stands.

The establishment of an adequate sampling protocol in the case of insect pests is important for the correct monitoring of the evolution of the damage caused [50]. The protocol involves identifying the presence of harmful insects based on field visits and visually assessing the level of damage in a subset of trees in both the crown and trunk [53]. In this study, monitoring protocols were created to observe the presence of larvae and oothecae and the degree of defoliation generated by *Gonipterus*. Monitoring is achieved if the same areas are inspected again, annually, or with a lesser frequency, using the same methods [53]. This work presents the frequency of visits and establishes a minimum of two visits annually with a fixed sampling period. Less intensive monitoring is implemented in Portugal [17], focusing on June. However, the number of trees and the evaluation of the damage by comparison with calibrated photographs was the same as in the proposed protocol, although different levels of damage were considered in the case of Portugal, grouped into only four categories. The larvae and oothecae were also counted in half the number of trees as in the protocol created in this work [17]. This protocol is therefore similar, although more restrictive than the one found in other works, and the methodology is repeatable in the affected areas.

The use of biological control with egg or larval parasitoids of the insect pest is one of the management strategies used to control the pest [50].

In Portugal, the losses produced in eucalyptus stands due to the pest over 20 years were evaluated, with the conclusion that early biological control is an effective action even though it may not lead to the disappearance of the defoliating insect [18]. A method has recently been devised to prepare risk maps to simulate the *G. platensis* pest in stands of *Spanish eucalyptus* spp. [42]. The results of these works are in agreement with the data on the development of the pest shown in this research.

Once the protocols for data collection and the application of parasitoids on the Cantabrian coast have been unified, it will be possible to calibrate the influence of other variables that could affect the results, such as temperature, stand density or altitude [17,42,54].

The results obtained for the biological control are evidence of the effectiveness of *A. nitens*, although it does not achieve total control of the pest. Therefore, it is necessary to study other parasitoids and their possible combinations [35,53] on the Cantabrian coast. Studies are currently underway in other countries on options such as the use of pheromone traps [30] and the deployment of *Steinernema diaprepesi* (an entomopathogenic nematode), which, due to its mutual association with bacteria, reproduces and kills the pupae of

*G. platensis* through septicaemia [31]. Additionally, two strains of entomopathogenic fungi have been discovered in Brazil that are especially lethal for adults of *G. platensis* [32].

Finally, forestry management could also alter the distribution of tree species and their pests [42]. The area occupied by *A. nitens* has increased substantially in the northern Iberian Peninsula due to its naturalisation in areas prone to frosts and its lower susceptibility to defoliation by *G. platensis* [55]. Changes in land use comprising an increase in urban and forested areas would affect the spread of pests [56]. In Spain, most plantations of eucalyptus spp. belong to small non-industrial private landowners, most of whom do not exercise the necessary management practices to obtain high productivity or in many cases, practice no active forest management at all [57]; however, owners' associations take pest control very seriously and develop and participate in the monitoring and control actions [7].

## 5. Conclusions

A unique protocol for evaluating damage by *G. platensis* would enable the harmonisation of detection systems, which could therefore be easily implemented by the AR and forest owners.

The optimised bioproduction of parasitoids (*A. nitens*) improves the biological control in the affected eucalyptus groves with a percentage of damage of less than 43%.

The degree of damage decreased compared to the data prior to the study. In Galicia and Asturias, there was a decline in the percentage of plots with a degree of defoliation over 1.

The pest monitoring frequency should be systematically amplified following the protocol, maintaining the sample size in order to adequately track its population dynamics and, therefore, the efficacy of the treatment.

**Author Contributions:** Conceptualisation, A.G.-I. and J.C.R.; methodology, E.A.-T. and J.C.R.; validation, E.A.-T., A.G.-I. and C.G.-G.; formal analysis, E.A.-T. and C.G.-G.; research, E.A.-T. and C.G.-G.; resources, A.G.-I. and J.C.R.; data cleansing, A.G.-I.; writing—original preparation of the draft, E.A.-T. and C.G.-G.; writing—revision and editing, E.A.-T. and J.C.R.; project administration, A.G.-I. and J.C.R.; fund acquisition, J.C.R. All authors have read and agreed to the published version of the manuscript.

**Funding:** This research was funded by the European Agricultural Fund for Rural Development (EAFRD) and the Ministry of Agriculture, Fishing and Food through the operational group GOSSGE with project identification number 20180020012387, part of the National Rural Development Programme (PNDR) 2014–2020 within the framework of the EIP-AGRI (European Association for Innovation in Productive and Sustainable Agriculture) to promote innovation in the agri-food and forestry sector: aid for the implementation of innovative projects of general interest or non-territorialisables developed by supra-regional operational groups.

**Institutional Review Board Statement:** Not applicable.

**Informed Consent Statement:** Not applicable.

**Conflicts of Interest:** The authors declare they have no conflict of interest.

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
