# Peer review of "Actions for Monitoring the Gonipterus Pest in Eucalyptus on the Cantabrian Coast"

_agronomy, doi:10.3390/agronomy12071692_

Round 1
Reviewer 1 Report
The present study provides a survey protocol for monitoring Gonipterus platensis, information about the level of infestation and defoliation of Eucalyptus trees from the insect and also some data about the presence and abundance of the insect in certain areas of Spain. Although the data are interesting my opinion is that the survey protocol for monitoring Gonipterus platensis is not clearly presented. The manuscript is written in a confusing way for the reader and must be improved.
General remarks
The introduction part is very poor. Please give more information about the life history of Gonipterus platensis, symptoms of infestation, distribution in Spain and in the EU and the methods that are generally used for monitoring this pest. Give more details about the biological control of the insect. Are there any other parasites or predators that can be used against G. platensis?
Materials and Methods must be improved and should include the methodology which was used to obtain the results regarding the adults, larvae and oothecae of the insect.
The results concerning the biological control are speculations. They are not supported by statistical analysis.
The discussion part is extremely limited. It must be enriched.
The term “plague” throughout the manuscript is not the appropriate. Use instead the name of the insect or “insect infestation”.
Some additional comments for the authors are given below.
Line 18: please correct “Curculionidae family”
Lines 31-45: give references for all the information that are given in this part.
Lines 73 -78: information about Anaphes nitens must be given in the introduction section. Those are not Material and Methods.
Line 76 and Line 78: replace “ootheca” with ”egg”
Lines 78 – 81: Please explain the term “bag”. What do you mean by “bags of oothecae”, “bags deposited with parasitised oothecae” and “number of oothecae per bag”?
Lines 90-99: did those data regard only to defoliation or there were and other additional information available?
Line 169-170: why did you make a break in July and August?
Figure 2: what do you mean by “Laying”?
Figure 7: how did you obtain the results about the adults of the insect? Did you use some kind of trap? Please give details in Materials and Methods.
Lines 254-258: how did you obtain the results about the larvae? Did you make samplings? Give details in Materials and Methods.
Line 258-259: this phrase must be in the previous paragraph since is related to adults.
Figure 8: how did you collect the oothecae? Give details in Materials and Methods.
Line 306: did you have any idea how many approximately were the parasitized eggs in each ootheca?
Lines 310-313: how did you reach to those conclusions? Is there any statistical analysis to support them?
Author Response
Thank you for your detailed review and feedback.
The response point-by-point is in the attached file.

Author Response

(The authors gave the same response as above.)

Reviewer 3 Report
This study aims to evaluate the survey of insect damage carried out within the Autonomous Regions (AR) and proposes a unified method and variables based on previous biological control campaigns survey data. This to monitor defoliation change caused by the eucalyptus snout beetle (Gonipterus platensis) and parasitism levels from Anaphes nitens. The results were analysed using the Statgraphics Centurion XVIII statistical software and used to assess summary statistics and LSD means to create confidence intervals and respective graphs.
The paper requires substantial revisions and improved clarity in the introduction, methods and results sections. Specific comments below.
Abstract
1. Please include more quantitative results information here of the study.
Introduction
2. Page 2 LINE 54. It would be good to mentioned the advantages of surveying such pests or their impacts, and what are some of the traditional approaches and then lead on to what has been done elsewhere as per the below comment.
3. Page 2 LINE 57. I suggest adding a few studies that have been done elsewhere, if not on this insect but a similar one. It would also add context to this study by showing what other methods and variables were used to survey similar types of plagues carried out by insects and what they have proposed.
Methods
4. Line 88. 2.3 Information used.
This section is disjoint. It would be better to explain either using text or a table what was received from each, and some basic information related to what data and variables and how many plots or samples etc.
5. Page 3 Line 118-131. “The minimum number of plots…..” I expected this text to be discussed/integrated under the study variables section and not the descriptive stats.
Results and Discussion
6. Line 147. Counts of trees? And how many plots is not clear? Is this related to table 2?
7. Page 4. 3.1. Survey protocol and 3.1.1. Sample selection should be discussed in the methodology section and not results. This should be discussed using text and not in point form.
8. The above comment applies to section 3.1.3. Oothecae collection.
9. The discussion is poor. “The results of these works are in line with the findings of this research.” This statement was made with no substantiation, discussion and reference to relevant material.
Author Response

(The authors gave the same response as above.)

Round 2
Reviewer 1 Report
The revised manuscript is improved.
Line 93: replace “ootheca” with “egg”.
Line 119: “Huber and Prinsloo” not italic
Line 125: delete “and”
Line 140: replace “Gonipterus platensis” with “G. platensis”
Line 144 -145: correct “Eucalyptus spp.”
Line 163: correct “from 26%”
Line 289: replace “lays” with eggs”
Figure 2: please explain the figure. How is it possible the egg laying period and the period of larvae development to be exactly the same? According to Lines 82 – 83 “the larvae hatch 10-15 days after laying”. Moreover, if laying of eggs and larvae development continue until December the insect must be able to hibernate at all stages (eggs, larvae, pupae and adults). Lastly, according to your figures 8 -11 no oothecae and larvae were observed in autumn of 2017, 2018 and 2019 which is contradictory to figure 2. Please clarify this point.
Author Response
The response to Reviewer 1 is in attached file.
